# Preparation of a CAB−GO/PES Mixed Matrix Ultrafiltration Membrane and Its Antifouling Performance

**DOI:** 10.3390/membranes13020241

**Published:** 2023-02-17

**Authors:** Haiyan Wu, Ling Wang, Wentao Xu, Zehai Xu, Guoliang Zhang

**Affiliations:** 1Center for Membrane and Water Science &Technology, State Key Laboratory of Green Chemical Synthesis Technology, Institute of Oceanic and Environmental Chemical Engineering, Zhejiang University of Technology, Hangzhou 310014, China; 2Hangzhou Special Equipments Inspection and Research Institute, Hangzhou 310005, China; 3College of Chemical Engineering and Material Science, Quanzhou Normal University, Quanzhou 362000, China

**Keywords:** antifouling, zwitterionic polymer, polyethersulfone (PES), graphene oxide (GO), mixed matrix ultrafiltration membrane

## Abstract

Serious membrane fouling has limited the development of ultrafiltration membrane technology for water purification. Synthesis of an ultrafiltration membrane with prominent anti-fouling ability is of vital importance. In this study, CAB−GO composite nanosheets were prepared by grafting graphene oxide (GO) with a zwitterionic material cocamidopropyl betaine (CAB) with strong antifouling properties. Anti-fouling CAB−GO/PES mixed matrix ultrafiltration membrane (CGM) was prepared by the phase inversion method with polyethersulfone (PES). Due to its electrostatic interaction, the interlayer distance between CAB−GO nanosheets was increased, and the dispersibility of GO was improved to large extent, thereby effectively avoiding the phenomenon of GO agglomeration in organic solvents. Based on the improvement of the surface porosity and surface hydrophilicity of the CAB−GO/PES mixed matrix membrane, the pure water flux of CGM−1.0 can reach 461 L/(m^2^·h), which was 2.5 times higher than that of the original PES membrane, and the rejection rates toward BSA and HA were above 96%. Moreover, when the content of CAB−GO was 0.1 wt%, the prepared CAB−GO/PES membrane exhibited very high BSA (99.1%) and HA (98.1%) rejection during long-term operation, indicating excellent anti-fouling ability.

## 1. Introduction

As a popular technology of membrane separation, ultrafiltration membrane technology is widely applied in water purification, food sterilization, bioengineering and desalination fields due to its high separation efficiency and low energy consumption [1,2,3,4]. Polyether sulfone (PES) has become an important membrane material for fabricating an ultrafiltration membrane due to its heat/oxidative endurance and excellent mechanical strength [5]. However, the membrane fouling phenomenon, especially the deposition of natural organic pollutants (NOM) including humic substances, proteins, peptides and polysaccharides [6] during the operation greatly limits the practical application of ultrafiltration membranes [7,8]. Serious concentration polarization is formed on the membrane surface, which leads to the degradation of membrane permeation, separation performance and service life [9,10,11,12]. Therefore, the elimination of membrane fouling is of vital importance.

The antifouling ability of ultrafiltration membranes can be improved by methods such as membrane surface coating and blending modification, thus improving the practical application of ultrafiltration membranes. Among them, blending modification by incorporating nanomaterials can not only make up for the defects of raw materials with modified materials, but also alleviate the membrane fouling phenomenon during the filtration process. Among numerous blended nanomaterials, graphene oxide (GO) nanosheets have attracted great attention as a good filler due to the high specific surface area, superior mechanical properties and their abundant hydrophilic groups [13]. The introduction of GO can effectively enhance the hydrophilicity and mechanical capacity of the blended membranes [14,15]. However, GO nanosheets are easily prone to agglomeration in organic solvents during the preparation of ultrafiltration membranes, resulting in the poor operating stability [16,17,18,19]. At the same time, larger NOM solutes are trapped and blocked on the membrane surface to form membrane fouling.

Zwitterionic polymers have strong hydrophilicity, and the anions and cations on the structure of zwitterionic materials are evenly distributed and equal in number, which can form a denser hydration layer than traditional hydrophilic materials [20,21]. In the case of strong hydration force, and strong electrostatic interaction, it is difficult for pollutants to be adsorbed on its surface. Among various zwitterionic polymer materials, the cocamidopropyl betaine (CAB) has strong hydrophilicity, and the compression of long alkane chains carried by CAB can cause steric repulsion, thus resisting the adsorption of NOMs such as protein on the surface [22]. With the aid of the hydrophobic carbon skeleton and abundant hydrophilic groups, the GO can provide abundant grafting active sites for zwitterionic molecules. Under the electrostatic interaction, the dispersibility of GO can be largely improved, thereby effectively improving the permeation separation efficiency and anti-fouling performance of the membrane. Inspired by these concepts, we put forward an idea where the combination of zwitterionic polymer CAB and GO nanosheets may be a feasible strategy to prepare CAB−GO fillers so as to incorporate with polymer membranes with ideal antifouling performance. There is no report on the synthesis of antifouling CAB−GO-based mixed matrix ultrafiltration membrane for water purification so far.

In this study, CAB−GO composite nanosheets were synthesized and introduced into polyethersulfone (PES) membranes to prepare a CAB−GO/PES mixed matrix ultrafiltration membrane (CGM) (Figure 1). This method contains three main points: (1) Compared with other reported coupling agents such as sodium persulfate, potassium persulfate and ammonium persulfate as an initiator would not only introduce other substances and participate in the reaction, but also provide free radicals to achieve a high initiation efficiency at the same temperature. (2) CAB−GO nanosheets with oxygen-containing and amino groups can make strong interactions with the polymer matrix through hydrogen bonds, electrostatic interactions and covalent bonds, therefore obtaining prominent compatibility and dispersity in blended membranes. (3) The anions and cations on the CAB structure are uniformly distributed and equal in number, thus forming a denser hydration layer than traditional hydrophilic materials. The pollutants are difficult to adsorb on the membrane surface, thereby improving the membrane’s anti-fouling performance. As a comparison, zwitterionic polymers with short chains were used to decorate GO nanosheets, the solvent dispersibility was also explored. The effects on surface morphology structure and property of membranes with different amount of CAB/GO were investigated, and the selective permeability of the mixed matrix ultrafiltration membranes were studied by water flux and BSA/HA rejection performance.

## 2. Experimental Section

### 2.1. Materials

All the chemicals and reagents used were analytical grade. *N*,*N*-dimethylacetamide (DMAc, 99.8%), *N*,*N*-dimethylformamide (DMF, 99.5%), methanol (99.9%), Cocamidopropyl Betaine (CAB, >95%), Polyethersulfone, Polyvinylpyrrolidone (PVP, Mw = 10,000), Humic Acid (HA, 99%), NaOH (97%), NaNO_3_ (99%), KMnO_4_ (99%), HCl (99.5%), H_2_O_2_ (30 wt%), H_2_SO_4_ (98%), Flake graphite (99.9%) and Bovine Serum Albumin (BSA, Mw = 66,000) were all purchased from Aladdin Co. China Ltd (Shanghai, China). Deionized water was obtained from a self-made two-stage RO device with a conductivity of 1.0–5.0 μS/cm.

### 2.2. Synthesis of CAB−GO Nanosheets

CAB−GO nanosheets were synthesized via an induction-grafting method by using (NH_4_)_2_S_2_O_8_ as initiator. First, highly oxidized GO nanosheets were prepared by the improved Hummers method [23]. A total of 1.5 g of flake graphite and 3 g of NaNO_3_ were added to concentrated sulfuric acid under ice bath, and then 6 g of KMnO_4_ was slowly added into the solution. After fully stirring at 35 °C for 3 h, water was then added dropwise at around 85 °C, and then the reaction system was transferred to oil bath at 98 °C for 0.5 h. Next, H_2_O_2_ was added and stirred with the mixture. The precipitate (graphite oxide) was obtained by centrifugation and washing, and the GO dispersion was obtained by sonication.

Then, 50 mL of DMF and 30 mL of 30 mg/mL CAB was added to 50 mL of 2 mg/mL GO dispersion, the solution was mixed and placed in an oil bath at 60 °C. Subsequently, 50 mL of 2 mg/mL (NH_4_)_2_S_2_O_8_ solution was added dropwise with a uniform rate under N_2_ atmosphere, and stirred steadily at 65 °C for 20 h in closed environment. The mixed solution was diluted with deionized water, and the remnant was removed. Finally, the CAB−GO nanosheets were obtained by washing and dried under vacuum at 70 °C. For comparison, GO was also grafted by glycine in a similar way and GLY−GO nanosheets were obtained.

### 2.3. Preparation of CAB−GO/PES Mixed Matrix Membranes

The CAB−GO/PES mixed matrix membranes were prepared by non-solvent phase separation. An appropriate amount of GO or CAB−GO was weighed and placed in a DMAc solution for ultrasonic exfoliation to obtain a uniform and stable GO−DMAc dispersion or CAB−GO/DMAc dispersion with a monolayer structure. A certain proportion of PVP was added and stirred at room temperature, then PES was added and fully stirred until the solid mixture was completely dissolved to obtain a homogeneous film casting solution. After standing and deaeration, a film scraper with a thickness of 200 nm was used to coat the surface of the quartz plate with the casting liquid at a constant speed. After letting it spread for 10–15 s, and the quartz plate loaded with the casting liquid was soaked in cold water (20 °C) to achieve phase separation. In order to avoid the occurrence of biological fouling and other conditions affecting the test results before the test analysis and characterization, the prepared membranes were placed in deionized water to remove the residual DMAc on the membrane and then all the prepared membrane samples were completely soaked in regularly replaced deionized water for later use. According to the different components and contents in the film, the film samples of each group were named with CGM-0, GOM, GGM, CGM-0.05, CGM-0.1, CGM-0.3, CGM-0.5 and CGM-1.0, respectively. (The configuration ratio of materials in each PES mixed matrix membranes were shown in Table 1).

### 2.4. Characterization

Fourier transform infrared spectroscopy (FTIR, Nicolet 6700, Thermos, Waltham, MA, USA), X-ray diffraction (XRD, Empyrean, Pnalytical, Almelo, The Netherlands) and X-ray photoelectron spectroscopy (XPS, Scientific K-Alpha, Thermo, Waltham, MA, USA) was used to characterize the surface functional groups, crystal structures and corresponding chemical states of the prepared GO and CAB−GO composites, respectively. Microstructures of the synthesized ultrafiltration membranes were characterized by scanning electron microscopy (SEM, Sigma 300, Zeiss, Birmingham, UK), and all the samples were gold-sprayed before SEM testing. The surface charge properties of CAB−GO composite membranes were analyzed by an electrokinetic analyzer for solid surface analysis (SurPass 3, Anton Paar, Graz, Austria) with solution of KCl. The hydrophilicity of each membrane was measured and compared by the contact angle meter (CA, OCAT21, Dataphysics, Filderstadt, Germany). All the samples were tested more than 3 times to reduce errors. The mechanical properties were evaluated by a double column high and low temperature tensile testing machine (5960, Instron, Norwood, MA, USA), and each membrane was tested at least 5 times to reduce errors.

### 2.5. Membrane Porosity and Average Pore Size

The average pore size of the prepared membrane was determined by the gravimetric method. After measuring the dry film mass (*ω*_0_) of each membrane samples, the membranes were soaked in water for more than 24 h to ensure complete wetting and steady state, then the wet weight (*ω*_1_) was recorded, and the membrane porosity (*ε*, %) was calculated. Combined with the membrane thickness, the corresponding average pore size (*r_m_*, m) was calculated. To ensure the accuracy of the data, 5 films were taken from each group for measurement, and each film was weighed 3 times and take the average value. The corresponding formulas for membrane porosity and the corresponding average pore size were as follows:$$\varepsilon \left(\%\right)=\frac{\omega_{1}-\omega_{0}}{\rho \times A\times l}\times 100\%$$
where *ω*_0_ is the mass of the dry film (g), *ω*_1_ is the mass of the wet film (g), *l* is the thickness of the membrane (m), *ρ* is the density of water (1 g/cm^3^) and *A* is the effective membrane area (m^2^).
$$r_{m}=\sqrt{\frac{\left(2.9-1.75\varepsilon \right)\times 8\eta lQ}{\varepsilon \times A\times \Delta P}}$$
where *η* is the viscosity of water (8.9 × 10^−4^ Pa·s), *Q* is the flow rate of pure water (m^3^/s), Δ*P* is the trans-membrane pressure (0.1 MPa).

### 2.6. Permeation and Separation Performance Experiments

Filtration experiments were performed in a laboratory-made cross-flow filtration unit (Figure 2) with BSA and HA as the main contaminants. After stabilizing at 2 bar operating pressure for 30 min, the pure water flux was tested at a steady pressure (1 bar), and then the feed solution was changed into BSA solution (0.2 g/L) or HA solution (0.02 g/L, pH = 7) [19]. The osmotic flux and the corresponding rejection rates of BSA solution (*J_B_*, L/m^2^/h/bar) or HA solution (*J_H_*, L/m^2^/h/bar) were recorded and all data were measured at 5-min intervals. The calculation formula of the pure water flux was below:$$J_{W}=\frac{V}{A\times \Delta t\times P}$$
where *J_w_* is the pure water flux (L/m^2^/h/bar), *V* is the permeation flux (L), *P* is the pressure through the membrane (bar), Δ*t* is the permeation time (h) and *A* is the effective area of the tested membrane (m^2^). The permeation fluxes of BSA (*J_B_*) and HA (*J_H_*) were also calculated by the above formula.

The contents of BSA and HA in the permeate were determined by UV-Vis spectrophotometer (UN-1102), and the rejection rates of BSA (*R_B_*, %) and HA (*R_H_*, %) were calculated by comparing the feed solution (*C*_0_, mg/mL) and the permeate (*C*_1_, mg/mL). The calculation formula of BSA separation performance (*R_B_*, %) is as follows:$$R\left(\%\right)=(1-\frac{C_{1}}{C_{0}})\times 100\%$$
where *C*_0_ and *C*_1_ are the contaminant concentrations (mg/mL) in the feed and permeate solutions and as measured by the wavelength of absorption maximum at 280 nm with a UV-Vis spectrophotometer, respectively. The rejection of HA (*R_H_*, %) was also calculated by this formula from the maximum absorption wavelength measured at 254 nm.

In addition, in order to determine the applicable transmembrane pressure range of ultrafiltration membranes, the permeation flux and separation performance of ultrafiltration membranes under different pressures were tested, and the cross-flow filtration device which was used in this study is shown in Figure 2. On the basis of a stable flow rate, the flux and rejection rate of the membrane under different transmembrane pressures (0.5 bar, 1 bar, 1.5 bar, 2 bar, 3 bar) were measured. The optimal transmembrane pressure of the ultrafiltration membrane was obtained by statistical analysis.

### 2.7. Mechanical Strength

The mechanical properties of the prepared membranes were tested by a high and low temperature double-column testing machine. All membranes were cut into rectangles of a fixed size (10 × 30 mm) and the thicknesses of the membranes were measured. All membranes were measured more than five times to reduce errors.

### 2.8. Antifouling Performance

In order to judge the anti-fouling property of the membrane, the membrane was subjected to a cycle test. Firstly, after measuring the flux of pure water (*J_w_*_,1_) for 30 min, the flux of BSA solution (*J_B_*) or HA solution (*J_H_*) was measured, and then the membrane was cleaned with distilled water at 2 bar for 0.5 h. Finally, the pure water flux (*J_w_*_,2_) of the stabilized membrane was tested at 1 bar [19]. The above steps were repeated three times, and the three-cycle experimental data were sorted and graphs were drawn. The flux recovery rate (*FRR*, %) is obtained by the following formula:$$FRR\left(\%\right)=\frac{J_{W,2}}{J_{W,1}}\times 100\%$$

### 2.9. Effect of Transmembrane Pressure on Membrane Performance

In order to determine the applicable transmembrane pressure range of ultrafiltration membranes, the permeation flux and separation performance of ultrafiltration membranes under different pressures were compared and judged. On the basis of a stable flow rate (25 mL/min), the changes in the permeability and rejection properties of the membrane under specific transmembrane pressures (0.5, 1.0, 1.5, 2.0 and 3.0 bar) were determined. The optimal transmembrane pressure of ultrafiltration membrane and the influence of transmembrane pressure on permeability of membranes were obtained.

## 3. Results and Discussion

To enhance the antifouling performance of PES ultrafiltration membranes, cocamidopropyl betaine (CAB) was used to modify the surface of graphene oxide for the first time, and the prepared mixed matrix ultrafiltration membranes with CAB−GO were used as additives. The poor compatibility and adhesion between GO and polymers lead to concomitant interfacial defects, which will reduce the permeation selectivity and mechanical properties of the composite membranes. The hydrophobic group of CAB can improve the compatibility of the optimized inorganic nanofiller (GO) with the polymer matrix (PES) [24]. As expected, after 4 h of static sedimentation, GO gradually began to show agglomeration and sedimentation (Figure 3a), and the sedimentation effect was more obvious after 24 h. On the contrary, the dispersibility of CAB−GO in DMAc was significantly improved by grafting CAB molecules, and it maintained excellent dispersibility over 24 h (Figure 3b), which effectively alleviated the agglomeration of GO nanosheets.

### 3.1. Structural Characteristics of GO and CAB−GO

In order to explore the differences of surface functional groups after grafting CAB, FTIR analysis was carried out. As depicted in Figure 4a, it was found that the absorption peak at 3374 cm^−1^ belonged to the –OH stretching vibration peak on the surface of GO, the absorption peak at 1225 cm^−1^ corresponded to the stretching vibration peak of the C–O single bond of GO and the absorption peak at 1054 cm^−1^ was the characteristic peaks of GO epoxy group. The absorption peaks at 1617 cm^−1^ and 1717 cm^−1^ corresponded to the stretching vibration of the unoxidized carbon-carbon double bond (C=C) and the C=O of the carbonyl and carboxyl groups at the edge of GO, respectively [25]. After the grafting modification of GO by CAB, the absorption peak of hydroxyl at 3374 cm^−1^ basically disappeared, and a new absorption peak of secondary amide appeared at 3278 cm^−1^. The absorption peak located at 1534 cm^−1^ and 1540 cm^−1^ was attributed to the –CO–NH− in-plane bending angle and –NH_2_– variable angle, respectively. Moreover, the peaks at 1550 cm^−1^, 2852 cm^−1^ and 2923 cm^−1^ corresponded to the –CH_3_ symmetrical deformation vibration, –CH_2_ symmetric stretch peak and –CH_2_ antisymmetric stretch peak [26,27]. The results demonstrated that the CAB molecules were successfully introduced in GO nanosheets.

The XRD pattern of GO (Figure 4b) is consistent with the diffraction patterns in other studies, and there was only one diffraction peak of GO at 12.1°, which indicates that the graphite was completely oxidized to obtain graphene oxide with a complete crystal structure. After the modification of GO with CAB, the diffraction peak shifted to the left and the intensity decreased greatly, proving that some oxygen-containing groups (hydroxyl, carboxyl, epoxy) were largely consumed during the modification process [28]. Since the GO nanosheets were partially reduced during the modification process, the quaternary ammonium salt groups on the surface in CAB grafted GO nanosheets reduced the integrity of the original crystal structure of GO, resulting in a smooth and broad camel shape appeared around the 18.8° peak, which conformed to the assumption in this study.

XPS analysis was used for the qualitative analysis of the structure and chemical states of the compound by exciting the valence electrons inside the molecule by X-rays. Figure 5a shows the XPS broad spectra of GO and CAB−GO. The specific performance was as follows: N 1s appeared at 399.08 eV in the spectrum of CAB−GO and the long chain of alkane (–C_12_H_25_) increased the C atom content in CAB−GO from 68.99% to 76.36% of GO. Due to the reduction in GO nanosheets during the modification process and the participation of some oxygen-containing functional groups in the chemical bonding process, the content of oxygen-containing functional groups was reduced, resulting in a decrease in the O content of CAB−GO from 29.38% before compounding to 19.3% [29]. Figure 5c shows the C 1s peaks of GO, in which the four peaks at 284.8 eV, 286.8 eV, 287.1 eV and 288.6 eV corresponded to C–C, C–O–C, C=O and COOH functional groups. Compared with GO, the C 1s peak of CAB−GO (Figure 5c) at 285.2 eV corresponded to the C−N bond was generated after recombination, and the N 1s peak of CAB−GO (Figure 5d) presents two different chemical peaks [30], which were attributed to the –NH–CO– (399.3 eV) and quaternary nitrogen (402.0 eV) in CAB, respectively, confirming the existence of amide groups and quaternary ammonium groups on the surface of GO [31,32]. All these phenomena indicate that CAB was successfully grafted on the surface of GO, which conformed to the FTIR results.

### 3.2. Structural Properties of CAB−GO/PES Mixed Matrix Membranes

To explore the effect of GO and CAB−GO on the microstructure of the PES membrane, SEM was used to characterize it, and the changes in the morphology and structure of the prepared membrane after modification by CAB−GO were further explored. Figure 6 shows the SEM images of GOM and CGM with different CAB−GO concentrations, including the membrane surface and cross section. The surface of each membrane was smooth without obvious differences. However, CGM-0.5 and CGM-1.0 had obvious micropores on the membrane surface. After the introduction of GO, some pore channels were blocked due to the π-π conjugation interaction of the GO nanosheets, which limited their compatibility and dispersibility (Figure 6c). The asymmetric membrane structure (sponge pore structure at the bottom of the membrane, the support layer finger pore structure at the middle position and the dense separation layer surface at the membrane surface) exhibited by introducing different amounts of CGM was similar to the PES ultrafiltration in other studies [33]. Given the presence of equal amounts of positive and negative charges on the surface of CAB, this would weaken the π-π interaction of GO nanosheets, and it exhibited a high affinity for DMAc and accelerated the diffusion of water and organic solvents, thereby accelerating the curing process of the membrane and forming larger pore channels [34]. Therefore, the size of the finger-like pores also enlarged to different extents by enhancing the content of CAB−GO [35]. Among them, the number of finger-like pore in the PES membrane substrate gradually decreased with the further increase in the CAB−GO content, while the numerous sponge-like pore structure was generated. Some micropores also appeared on the membrane surface of CGM-0.5 and CGM-1.0. This was attributed to the fact that CAB−GO increased the viscosity of the CGM solution, which limited the bidirectional diffusion behavior (the non-solvent phase and the solvent phase of the solution) and adversely affected the phase inversion rate, which finally changed the microscopic pore structure of the CGM under this dual action [36]. Obviously, the content of CAB−GO has a remarkable impact on the microscopic pore structure of the mixed matrix ultrafiltration membranes.

It can be seen from Figure 7 that the surface of CGM-0 was electronegative in a relatively wide pH range of 4–10, which made it easier to adsorb positively charged pollutants and deposit them on the membrane surface due to sieving and electrostatic interaction during filtration. After the incorporation of GO, a higher zeta potential was shown on the surface of GOM, because GO carried a large number of carboxyl groups, which ionized the hydrogen ions in the solution, thus endowing GOM with a certain electronegativity (the negative surface charge depends on the number of carboxyl groups per unit surface area, which is the carboxyl group density) [37,38]. Obviously, the PES mixed matrix membrane with CAB−GO as the modification additive has both positively and negatively charged components, which can be uniformly distributed on the membrane surface and become a zwitterion-like material [39]. Since the positively charged ions carried by CAB are relatively weak, the zeta potential of the membrane surface gradually decreases with the increasing of pH [40].

### 3.3. Mechanical Properties

Mechanical strength specifically includes elongation at break, elastic modulus, tensile strength and other major parameters, which can be used to evaluate the practicality of mixed-matrix ultrafiltration membranes. Table 2 shows the mechanical properties of pure PES, GOM, GGM and different CGM and the results indicate that the PES mixed matrix membranes blended by GO and its derivatives improved the mechanical properties of PES membranes. Among them, the tensile strength and elastic modulus of GOM were increased from 1.43 MPa (CGM-0) to 1.76 MPa, and the elastic modulus were increased from 68.61 MPa to 81.61 MPa, respectively. Tensile strength of GGM was increased to 1.99 MPa and the elastic modulus of 103.14 MPa, indicating that the functional group of GO enhanced the interface interaction between GO and polymer, so the incorporation of GO improved the mechanical strength of the mixed matrix membrane. CGM-0.1 showed a tensile strength of 1.47 MPa and an elastic modulus of 88.43 MPa. Although it was slightly lower than GGM, it exhibited an elongation at break (15.75%) better than that of GOM and GGM. By further increasing the content of CAB−GO, the tensile strength enhanced from 1.47 MPa to 2.04 MPa, and the elongation at break also enhanced from 15.75% to 17.02%. The fundamental reason is that CAB−GO contains a large number of functional groups, which can effectively enhance the interaction between the membrane substrate and the CAB−GO interface. Therefore, CAB−GO is more easily highly dispersed into the PES membrane matrix, so improving the mechanical stability of the blended membranes. The stacking of CAB−GO led to a decrease in compressive strength, while the CAB−GO concentration exceeded 0.3 wt%, resulting in the decrement of elongation at break. Moreover, with the increase in the content of CAB−GO, the elastic modulus of the membrane gradually decreased from 88.43 MPa to 65.18 MPa, which was mainly because the fillers appeared in different degrees of stacking in the polymer matrix, which gradually weakened the force between the fillers and the polymer.

### 3.4. Hydrophilic Properties

Recent studies have shown that the hydrophilic property of the membrane is an important factor affecting its anti-fouling performance. Compared with hydrophobic contaminants (HA, BSA), the highly hydrophilic membrane surface preferentially binds to water molecules during filtration, which can effectively avoid the partial attachment of organic contaminants [41]. Figure 8 shows the water contact angle of CGM-0, GOM, CGM-0.05, CGM-0.1, CGM-0.3, CGM-0.5 and CGM-1.0, respectively. After incorporation of CAB−GO, the water contact angle of the PES film was greatly reduced. As the CAB−GO content in the CGM casting solution increased, the contact angle gradually decreased from 86.4° to 72.4°, which was caused by the hydrophilic groups in CAB−GO improving the hydrophilicity of the CGM membrane. In the process of increasing CAB−GO addition, the water contact angle was slightly changed. The experimental results indicate that the hydrophilic performance of the PES ultrafiltration membrane significantly improved after modification, and the corresponding pure water permeability and antifouling performance also showed a high level.

The pore structure parameters of different membrane surfaces measured by gravimetry are shown in Table 3. Both the modified membranes had a higher porosity and average pore size than the membranes without modifiers. It can be noticed that after adding GO to the PES matrix, the porosity increased by 2% and the average pore size increased by 1.6 nm, indicating that adding hydrophilic GO can enhance the thermodynamic instability of the casting solution, thereby shortening the liquid–liquid stratification delay time during the phase separation process, achieving better porous structures. The porosity and average pore size of the membranes gradually improved when the amount of CAB−GO addition increased. When the addition of CAB−GO reached 0.5 wt%, the porosity and average pore size of the membrane reached their maximum values, which were 63.44% and 14.73 nm, respectively. Because of the stronger hydrophilicity, which is beneficial to the DMAc solution into the pure water, the CAB−GO can move rapidly to the membrane surface during the phase separation, increasing the porosity and average pore size of the ultrafiltration membrane [42]. The porosity of CGM-1.0 decreased slightly (63.00%) due to the increase in solution viscosity, while the addition of CAB−GO exceeded 0.5 wt% [43].

### 3.5. Permeability and Separation Performance

The permeation flux of the membrane is closely related to the membrane pore structure and surface hydrophilicity. Figure 9 shows that the pure water fluxes of CGM-0, GOM, GGM, CGM-0.05, CGM-0.1, CGM-0.3, CGM-0.5 and CGM-1.0 were 181.9, 280.6, 335.2, 302.74, 331.7, 367.3, 423.6 and 461 L/(m^2^·h), respectively, indicating that the water flux of the membranes blended by CAB−GO improved to varying degrees, and reached to the maximum when the content of CAB−GO nanosheets was 1.0 wt%. As depicted in Figure 8, the hydrophilic performance of PES ultrafiltration membrane was consistent with the change rule of its pure water flux when the content of CAB−GO nanosheet increased. This was mainly attributed to that abundant hydrophilic groups on the membrane surface easily adsorbed water molecules in the process of membrane separation and subsequently formed a hydration layer, thus allowing water to preferentially move across the membrane matrix, thereby improving the water flux of the membrane.

The rejection rate and the permeation flux of the membrane are closely related to the water quality effect after the filtration experiment, which are important indicators to characterize the properties of the membrane. Figure 9 shows that the membrane permeability indicates the filtration efficiency of the membrane, and the filtration quality of the membranes depends on the rejection performance [44]. Obviously, the introduction of CAB significantly improved the rejection rates of BSA and HA, effectively preventing the passage of pollutants while ensuring the water flux. When the loading amount of CAB−GO was 0.1 wt%, the as-prepared membrane had a rejection rate of 96.6% for BSA and 97.7% for HA. The rejection effect of BSA/HA was not only determined by the size sieving and hydrophilic properties of the membrane, but also influenced by electrostatic repulsion. The rejection rates of BSA from CGM-0.05 to CGM-1.0 were 94.2%, 96.6%, 93.7%, 91.8% and 87.8%, respectively. The rejection rates of HA were all higher than 90%, especially CGM-0.1 reached 97.7%, followed by CGM-0.05 and CGM-0.3, which reached 96% and 95.9%, respectively. The rejection rates of HA were higher than that of BSA because the size of the HA molecule was much larger than that of BSA. With the increase in CAB−GO ratio, the rejection rate of ultrafiltration membrane decreased. The CAB−GO blended mixed-matrix ultrafiltration membrane showed a higher HA separation performance; it benefited from the existence of abundant quaternary ammonium and hydroxyl groups in CAB−GO, which improved the rejection performance of the ultrafiltration membrane [36]. In addition, the pure water flux of GGM was slightly different from that of CGM-0.1, and there was a partial error overlap, which is consistent with the corresponding porosity and pore size distribution results (Table 3).

Figure 10 shows the effect of different operating pressures on the pure water flux, BSA/HA rejection of GOM and CGM. Membrane rejection tends to decrease with increasing transmembrane pressure because concentration polarization and fouling are more severe at higher transmembrane pressures, thus forming an additional selective layer on the membrane surface, which reduces the rejection rate [41]. The rejection rate of CGM first increased and then decreased during the increase in transmembrane pressure and showed the best rejection effect at 1.5 bar (BSA 99.1%, HA 98.1%). Due to the hydrophilic layer formed by CAB on the membrane surface, CGM-0.1 had excellent anti-fouling ability, and still maintained 94.6% BSA rejection and 96.4% HA rejection under the operating pressure of 3 bar, showing a strong stability.

### 3.6. Antifouling Performance and Stability

The water flux recovery rates (FRR) of different membranes are shown in Figure 11. The BSA FRR of the PES membrane was 32.3% and the HA FRR was 38.2%, which increased to 64.2% and 68.8% when 0.1 wt% GO was added. As the hydrogen bonds are relatively easy to break and recombine, GO as a hydrophilic material typically undergoes a transition from non-fouling to fouling upon changes in surface hydration caused by increasing bulk density [22]. Due to the strong hydrophilicity of the zwitterion, it exhibits a high FRR and promotes the formation of a hydrated layer, which can effectively prevent the sedimentation and adsorption of BSA and HA. The anti-fouling effect of the zwitterion-modified ultrafiltration membranes was better than GOM. Among them, the BSA FRR and HA FRR of GGM was 84.2% and 89.3%, respectively. The BSA FRR and HA FRR of CGM-0.1 was as high as 96.8% and 97.1%, respectively. This excellent antifouling capacity is due to the continuous compact hydration layer composed on the membrane surface by zwitterions through electrostatic interactions and steric hindrance effects [45]. Usually, high hydrophilicity always goes along with better antifouling properties, but this was not the case here, due to factors such as steric repulsion effects or the complexity of membrane morphology after modification [46]. As the hydrophilicity increases, a filter cake layer will gradually form on the membrane surface to resist the pollutants, thereby reducing the pollutants in the pores and affecting the permeate flux of the membrane.

The gradual decrease in permeate flux during membrane filtration was mainly caused by NOM, and three cyclic filtration processes were performed by using BSA or HA solutions as fouling sources to investigate the anti-fouling performance of the prepared membranes (Figure 12). Membrane flux decays over time because lots of solid contaminants accumulate on the membrane surface, where they are compressed and form a gel layer that blocks pores. However, a high permeation flux can still be maintained after hydraulic flushing. This behavior can be attributed to the hydrophilicity and tunable pore size, which reflected the better anti-fouling performance of the modified membrane against BSA and HA. Compared with CGM-1.0, which maintained 73.8% and 82.4% FRR in the three cycles of BSA and HA, respectively, CGM-0.1 showed a more stable penetration effect, which maintained 79.4% of BSA FRR and 82.9% of HA FRR in three cycles. This showed that CGM-0.1 has an outstanding anti-fouling performance and stability in continuous ultrafiltration process.

## 4. Conclusions

In order to improve the anti-fouling performance of a PES ultrafiltration membrane, CAB was used to graft GO for the first time, and the CAB−GO composite was introduced into a PES mixed matrix ultrafiltration membrane by the phase inversion method. The chemical structure and surface functional group distribution of CAB−GO were analyzed. The microstructure, hydrophilic properties, mechanical strength and surface chargeability of a CAB−GO/PES mixed matrix membrane were evaluated; The permeation and separation properties of CAB−GO /PES mixed matrix membranes were investigated, and the anti-fouling properties of membranes and the effects of different transmembrane pressures on membrane properties were systematically studied. The main conclusions are as follows:(1)The dispersibility of GO grafted with CAB was obviously better than that of pristine GO, and the dispersion effect could be maintained for up to 24 h, indicating that CAB provided GO with sufficient long alkane chains, quaternary nitrogen atoms and amide groups. Due to its electrostatic interaction, the interlayer distance between CAB−GO nanosheets was increased, and the dispersibility of GO was improved to large extent, thereby effectively avoiding the phenomenon of GO agglomeration in organic solvents;(2)Based on the improvement of the surface porosity and surface hydrophilicity of the CAB−GO/PES mixed matrix membrane, the pure water flux of CGM-1.0 reached 461 L/(m^2^·h), which was 2.5 times higher than that of the original PES membrane. The CGM-0.1 also had a high pure water flux which was 180% higher than the original membrane. The rejection rates toward BSA and HA were above 96%;(3)Humic acid (HA) and bovine serum albumin (BSA) were used as target pollutants, the BSA rejection rate and corresponding HA rejection rate were increased from 87.46% to 96.57%, and from 88.64% to 97.70% after introducing CAB−GO, respectively. In the process of increasing the transmembrane pressure, CGM-0.1 exhibited better BSA (99.1%) and HA (98.1%) rejection at 1.5 bar.

## Figures and Tables

**Figure 1 membranes-13-00241-f001:**
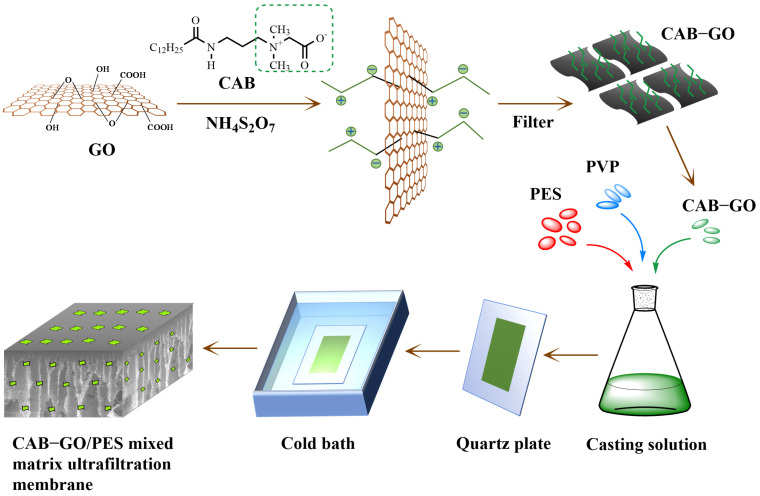
Scheme of the preparation process of a CAB−GO/PES membrane.

**Figure 2 membranes-13-00241-f002:**
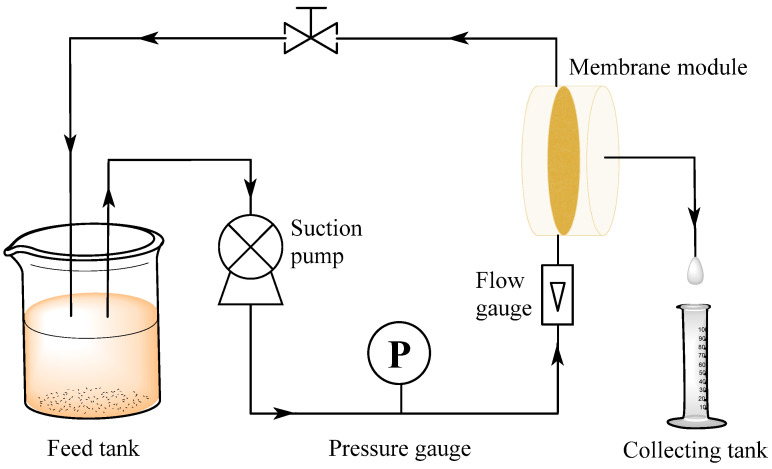
Scheme of laboratory homemade cross-flow filtration equipment.

**Figure 3 membranes-13-00241-f003:**
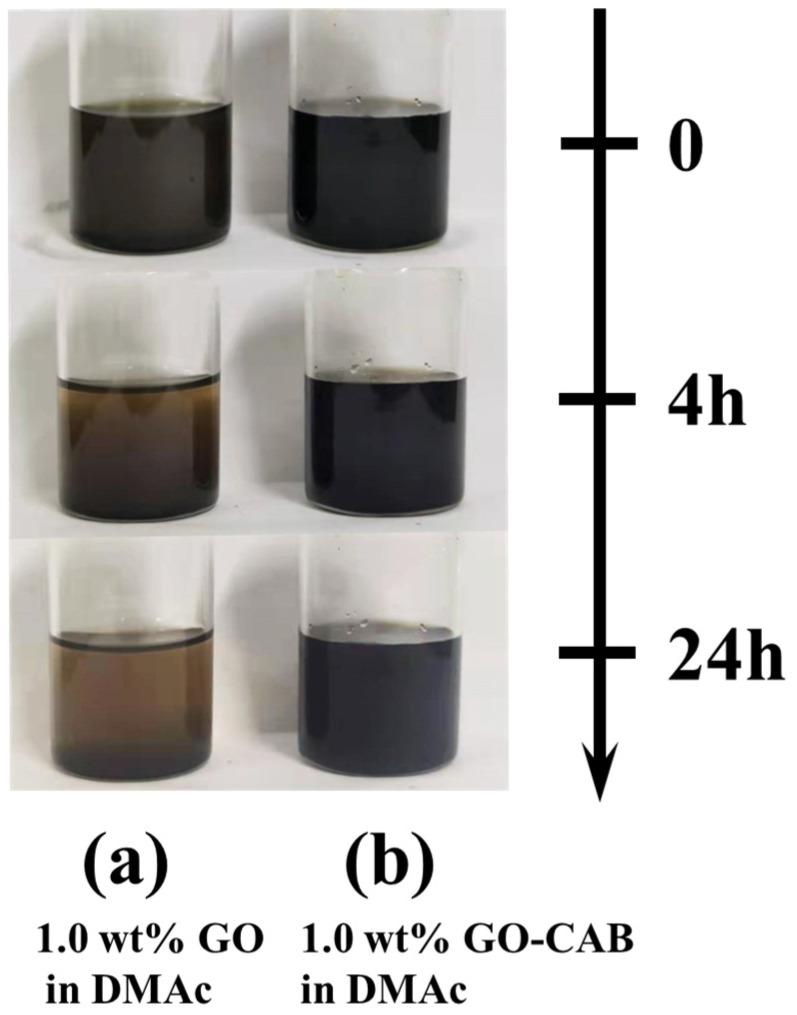
Photographs of 1.0 wt% GO (**a**) and 1.0 wt% CAB−GO (**b**) in DMAc.

**Figure 4 membranes-13-00241-f004:**
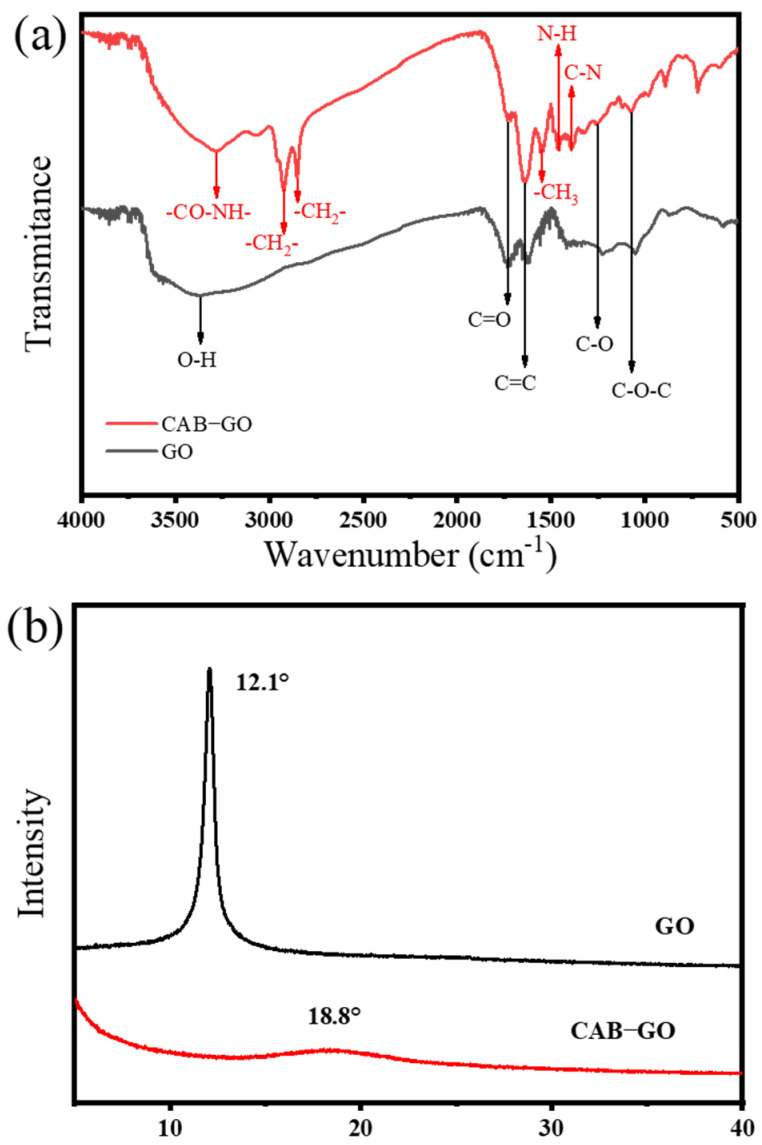
FTIR spectra (**a**) and XRD patterns (**b**) of GO and CAB−GO samples.

**Figure 5 membranes-13-00241-f005:**
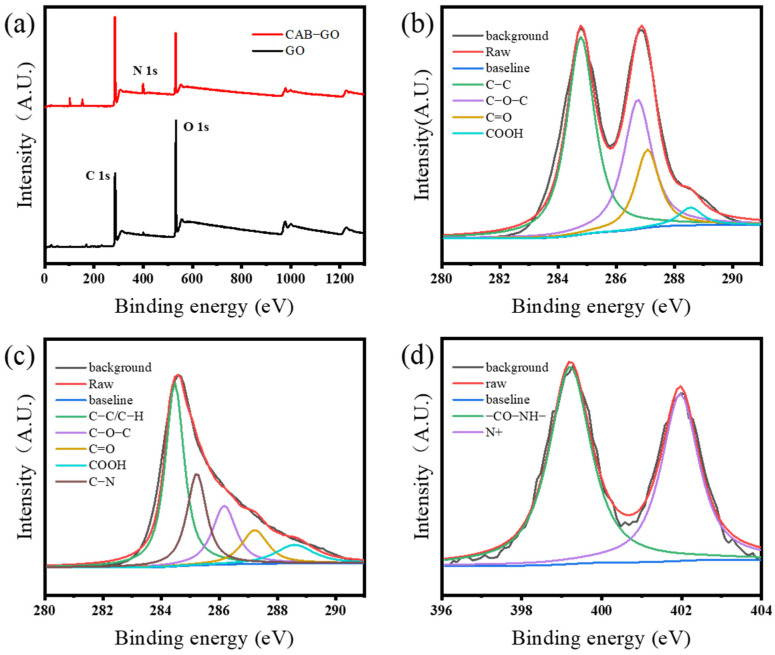
XPS full spectra of GO and CAB−GO (**a**), C 1s spectrum of GO (**b**), CAB−GO (**c**) and N 1s spectrum of CAB−GO (**d**).

**Figure 6 membranes-13-00241-f006:**
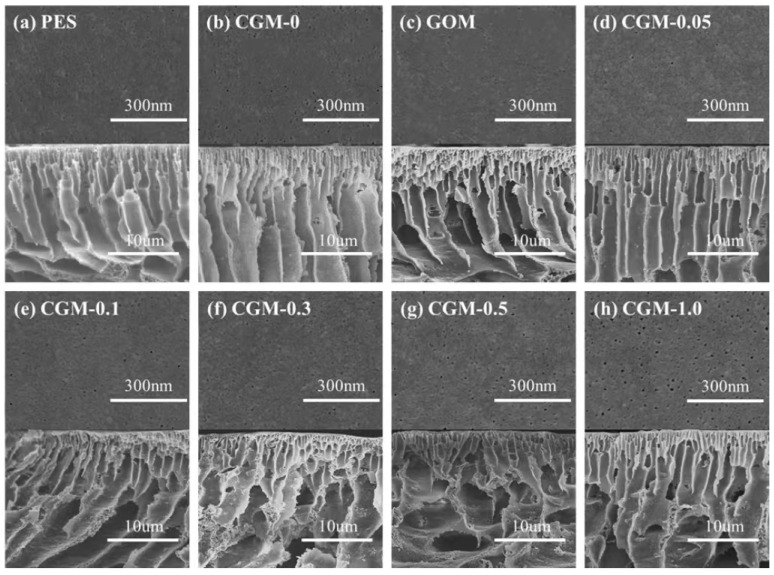
Surface and cross-sectional SEM images of pure PES (**a**), CGM-0 (**b**), GOM (**c**), CGM-0.05 (**d**), CGM-0.1 (**e**), CGM-0.3 (**f**), CGM-0.5 (**g**) and CGM-1.0 (**h**).

**Figure 7 membranes-13-00241-f007:**
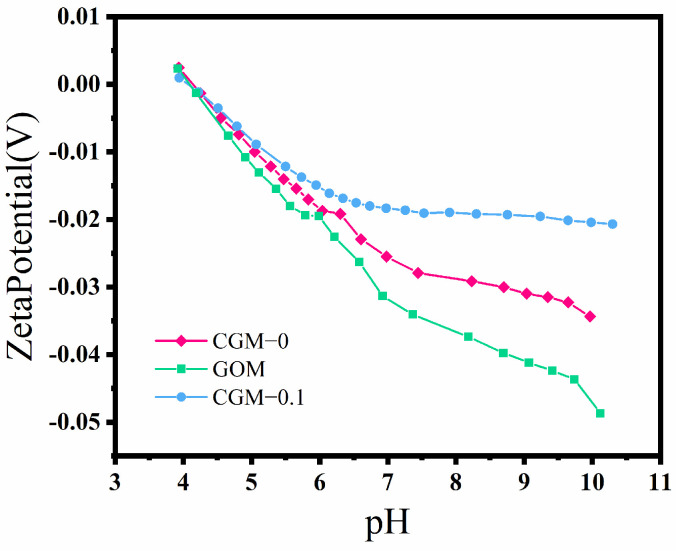
Zeta potential of CGM−0, GOM and CGM−0.1.

**Figure 8 membranes-13-00241-f008:**
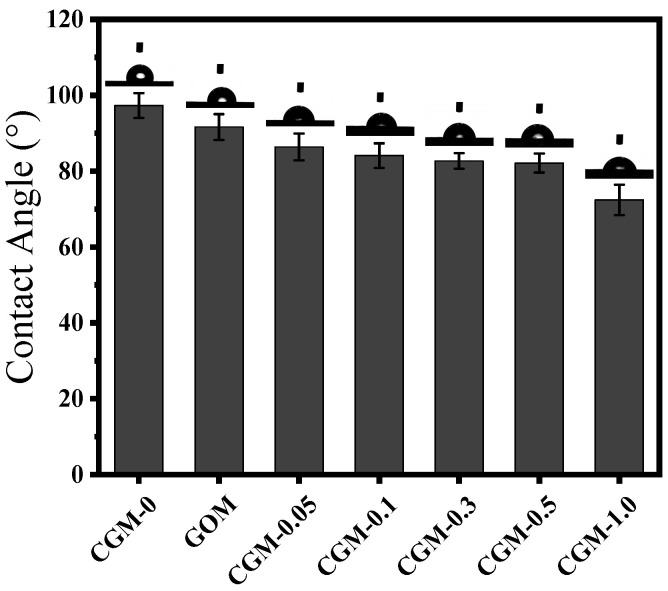
Water contact angle of different membranes.

**Figure 9 membranes-13-00241-f009:**
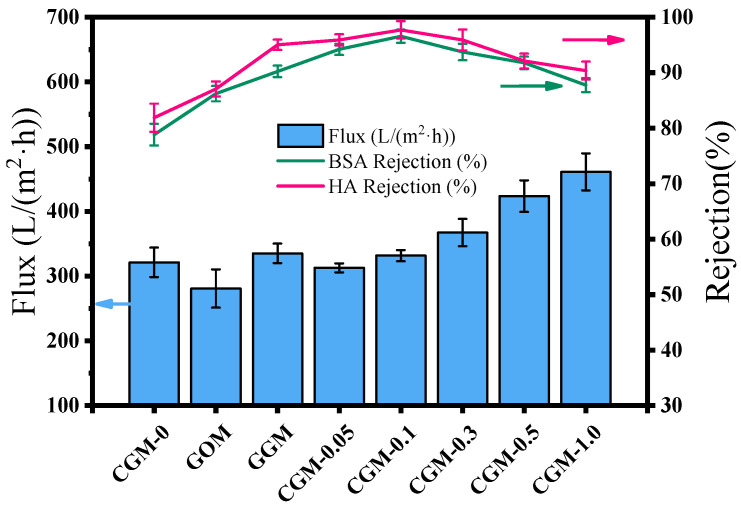
Pure water flux, BSA rejection and HA rejection of different membranes.

**Figure 10 membranes-13-00241-f010:**
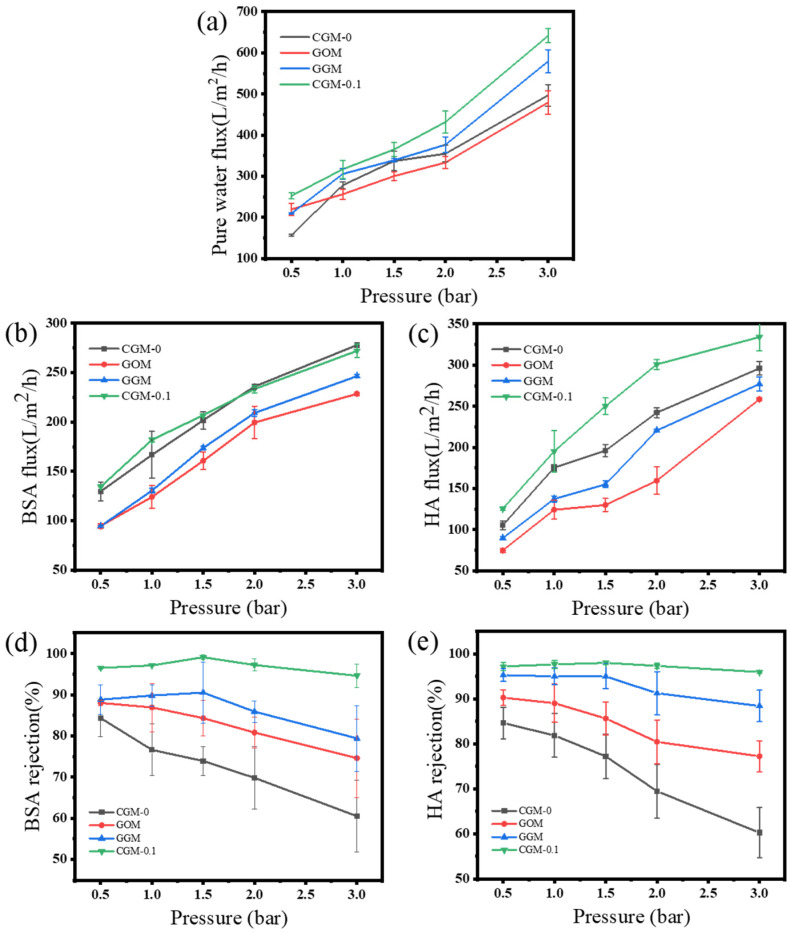
Pure water flux (**a**), BSA flux (**b**), HA flux (**c**), BSA rejection (**d**), and HA retention (**e**) for CGM-0, GOM, GGM and CGM-0.1 at different operating pressures.

**Figure 11 membranes-13-00241-f011:**
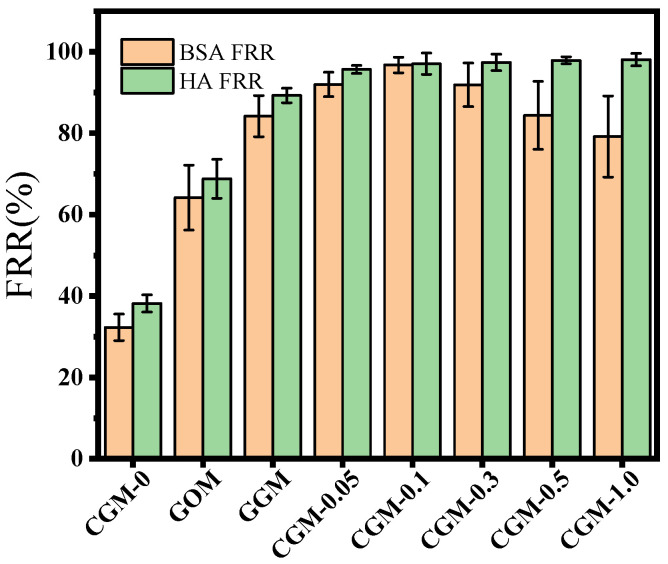
Flux recovery rate of different membranes in one cycle.

**Figure 12 membranes-13-00241-f012:**
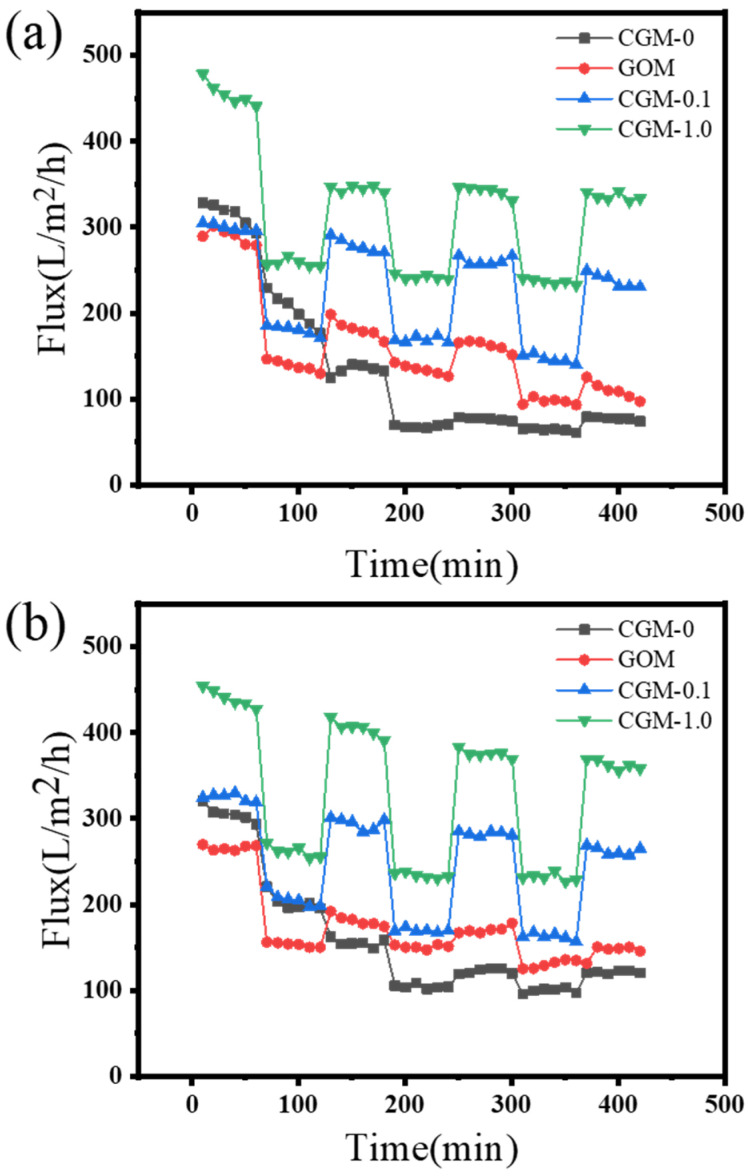
Water flux recovery of CGM-0, GOM, CGM-0.1 and CGM-1.0 after 3 cycles of BSA solution (**a**) and HA solution (**b**).

**Table 1 membranes-13-00241-t001:** The configuration ratio of mixed matrix membranes.

Membrane	PES (wt%)	PVP (wt%)	GO (wt%)	Gly−GO (wt%)	CAB−GO (wt%)	DMAc (wt%)
CGM-0	16	0.1	\	\	\	83.9
GOM	16	0.1	0.1	\	\	83.8
GGM	16	0.1	\	0.1	\	83.8
CGM-0.05	16	0.1	\	\	0.05	83.85
CGM-0.1	16	0.1	\	\	0.1	83.8
CGM-0.3	16	0.1	\	\	0.3	83.6
CGM-0.5	16	0.1	\	\	0.5	83.4
CGM-1.0	16	0.1	\	\	1.0	82.9

**Table 2 membranes-13-00241-t002:** Comparison of mechanical strength of ultrafiltration membranes.

Membrane	Tensile Strength (MPa)	Elasticity Modulus (MPa)	Breaking Elongation (%)
CGM-0	1.43 (±0.38)	68.61 (±11.29)	6.40 (±0.03)
GOM	1.76 (±0.27)	81.61 (±21.60)	4.20 (±0.39)
GGM	1.99 (±0.15)	103.14 (±5.34)	4.24 (±0.16)
CGM-0.1	1.47 (±0.44)	88.43 (±16.00)	15.75 (±1.77)
CGM-0.3	1.93 (±0.53)	83.08 (±8.33)	17.02 (±4.85)
CGM-0.5	2.04 (±0.87)	69.36 (±9.86)	10.23 (±2.58)
CGM-1.0	0.96 (±0.44)	65.18 (±8.20)	14.96 (±4.75)

**Table 3 membranes-13-00241-t003:** Porosity and average pore size of different ultrafiltration membranes.

Membrane	Overall Porosity (%)	Mean Pore Size (nm)
CGM-0	45.58 (±1.53)	8.15 (±0.34)
GOM	47.93 (±1.69)	9.75 (±0.63)
CGM-0.1	59.59 (±2.33)	10.70 (±0.79)
CGM-0.3	62.68 (±1.47)	13.55 (±0.51)
CGM-0.5	63.44 (±1.08)	14.73 (±0.73)
CGM-1.0	63.00 (±1.52)	14.90 (±0.58)

## Data Availability

Not applicable.

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
