# Peer review of "Preparation of a CAB−GO/PES Mixed Matrix Ultrafiltration Membrane and Its Antifouling Performance"

_membranes, 2023, doi:10.3390/membranes13020241_

Round 1

Reviewer 1 Report

Reducing membrane surface contamination is a current challenge in water treatment technology. The results obtained by the authors may find application in practice. However, there are some remarks to the work:

1. The purity of the materials used is not indicated.

2. What are the characteristics of deionised water?

3. Formulas for determining porosity and pore size should be detailed or references given.

4. What method was used to determine the density of the material?

5. What were the experimental conditions for determining porosity and pore size (temperature)?

6. What is the reason there is no pressure in the dimension for the flow?

7. Give a more detailed explanation of the observed non-linearity of the elastic-strength characteristics.

8. How was the contact angle of wetting determined?

9. References are not displayed correctly.

Reviewer 2 Report

Zhang et al. report the preparation of an anti-fouling CAB-GO/PES mixed matrix ultrafiltration membrane. They have used pure water to investigate the developed membrane and BSA and HA rejection during long-term operation to indicate the membrane’s anti-fouling ability.  The authors need to address the following before any possible publications:

1.       Revise the introduction by correcting the inserted citations, e.g., in lines 41, 51, 53, 57, …, 289, 313, .. etc. Also, nothing was mentioned about using BSA and HA.     

2.       Under the Experimental section: correct the dry/wet film mass and density symbols to match the corresponding formulas and insert relevant references for these and other formulas.

3.       Why the mechanical strength of membranes with different CGM (Table 1) is inconsistent?  

4.       The variation of the anti-pollution performance and stability of the membranes with different wt% GO should be explained. For instance, why does Fig. 8 show no differences in the HA rejection on all membranes with different wt% GO? However, the membranes vary in Flux recovery (Fig. 10e).

5.       The quality of all figures should be significantly improved in the revised manuscript, particularly Fig. 6. and Fig. 8 

Reviewer 3 Report

In the present work, graphene oxide (GO) impregnated with cocamidopropyl betaine (CAB) was produced and used in the manufacture of composite membranes with polyethersulfone (PES), aiming at an improvement of anti-fouling properties. From my point of view, the paper has an adequate theoretical framework, the methodology is well presented, and the results are properly discussed. Thus, I agree that the theme proposed and the results presented are valuable contributions to the literature. Nevertheless, some aspects need to be improved before the article is published.

General Comments and Suggestions for Authors:

1) Lines 121 - 124: “A film scraper with a thickness of 200 nm was used to coat the surface of the quartz plate with the casting liquid at a constant speed, and the quartz plate loaded with the casting liquid was soaked in cold water to achieve phase separation.” Was the casting liquid immediately soaked in cold water or was it left for some time to evaporate after spreading? What was the evaporation time (if it was performed)? Would it be possible to inform the temperature of the cold water?

2) Lines 129 - 130: I recommend informing the composition of the membranes (CGM-0, GOM, GGM, CGM-0.05, CGM-0.1, CGM-0.3, CGM-0.5, and CGM-1.0) in a Table. This will make the article easier to read and interpret.

3) Lines 138 - 139: Please inform which equipment and how the zeta potential analyzes of the membranes were carried out. If necessary, use references.

4) Lines 154 – 161: The description of some "variables" used in the formulas for calculating porosity and pore radius is incorrect. I recommend a careful review. W, δ, and ∆? are used in the equation, while ω, l, and P are used in the text, respectively. Also, check the use of the variable ?. Furthermore, on line 160 it says that "Q is the flux of pure water (m3/s)"; however, m3/s is a unit of flowrate and not flux.

5) In line 171 the following sentence is used: "...Jw is the pure water flux (L/m2/h)...". However, in the equation of line 170, Jw= ?/(? × âˆ†? × ?). There is some inconsistency here. According to the literature in the area of membranes, when the Flux (J) is divided by pressure (P) it originates a parameter known as Hydraulic permeability, water permeability, or permeance and has units of L/m2/h/bar). So I recommend correcting the current text and/or fitting the equation to the text.

6) The authors use the term "Anti-pollution" to refer to the antifouling property of the membranes. I understand that many times the researchers try to alternate words with their synonyms, however, the technical term used in the field of membranes is "fouling" and "antifouling". Therefore, I recommend these changes throughout the manuscript.

7) In the caption of Figure 3, add the concentrations of GO and CAB-GO used in the experiment that resulted in the photo.

8) Regarding to Figure 6. Practically nothing can be seen in the images of the surface of the membranes. They contribute little to the article. It would be better if images with a higher magnification were added.

9) Materials with a contact angle less than 90° are considered hydrophilic, but to be considered super-hydrophilic the contact angle must be close to 0°, usually less than 10°. Therefore, I recommend revising the following sentence (lines 419-420): "Due to the super-hydrophilic layer formed by CAB on the membrane surface….".

10) Review the caption of Figure 10, as it does not agree with the graphs presented.

11) Finally, the manuscript must be checked for typos:

Line 46: “…. materials, but also. Alleviate the…”.

Line 259: “Figure 5a. shows the ….”

Line 271: “…different chemical peaks 30. which were a …”

And so on…

Round 2

Reviewer 1 Report

Work has improved, but some questions remain unanswered. The formula for determining the pore size is not explained. In [19] it is also given without explanation. Formula for determination of porosity should be written in the classical form, as the ratio of pore volume to the sum of the volume of the material frame and the pore volume. The density of the framework material is determined by standard methods, e.g. helium pycnometry. Otherwise, reference should be done to the product data sheet.

Reviewer 2 Report

Accept in present form.

Author Response

It's ok.

Reviewer 3 Report

The authors addressed all comments and suggestions; however, I have found a minor point that needs to be revised. 

On line 164 the equation for calculating rm is presented. The variables "l" and "δ" appear in the equation, and in the text just below the equation, the variable "l" is described as the thickness of the membrane. So, what would be the "δ" of this equation? It looks like the "δ" is left over in the equation. Please check this out.

I think that this minor correction can be performed at the proofs step. Therefore, I recommend this valuable paper for publication in Membranes.
